# Randomized Prior Functions
# for Deep Reinforcement Learning

**Ian Osband**
DeepMind
iosband@google.com

**John Aslanides**
DeepMind
jaslanides@google.com

**Albin Cassirer**
DeepMind
cassirer@google.com

## Abstract

Dealing with uncertainty is essential for efficient reinforcement learning. There is a growing literature on uncertainty estimation for deep learning from fixed datasets, but many of the most popular approaches are poorly-suited to sequential decision problems. Other methods, such as bootstrap sampling, have no mechanism for uncertainty that does not come from the observed data. We highlight why this can be a crucial shortcoming and propose a simple remedy through addition of a randomized untrainable 'prior' network to each ensemble member. We prove that this approach is efficient with linear representations, provide simple illustrations of its efficacy with nonlinear representations and show that this approach scales to large-scale problems far better than previous attempts.

## 1 Introduction

Deep learning methods have emerged as the state of the art approach for many challenging problems [30, 70]. This is due to the statistical flexibility and computational scalability of large and deep neural networks, which allows them to harness the information in large and rich datasets. Deep reinforcement learning combines deep learning with sequential decision making under uncertainty. Here an agent takes actions inside an environment in order to maximize some cumulative reward [63]. This combination of deep learning with reinforcement learning (RL) has proved remarkably successful [67, 42, 60].

At the same time, elementary decision theory shows that the only admissible decision rules are Bayesian [12, 71]. Colloquially, this means that any decision rule that is not Bayesian can be improved (or even exploited) by some Bayesian alternative [14]. Despite this fact, the majority of deep learning research has evolved outside of Bayesian (or even statistical) analysis [55, 32]. This disconnect extends to deep RL, where the majority of state of the art algorithms have no concept of uncertainty [42, 41] and can fail spectacularly even in simple problems where success requires its consideration [40, 45].

There is a long history of research in Bayesian neural networks that never quite became mainstream practice [37, 43]. Recently, Bayesian deep learning has experienced a resurgence of interest with a myriad of approaches for uncertainty quantification in fixed datasets and also sequential decision problems [29, 11, 20, 47]. In this paper we highlight the surprising fact that many of these well-cited and popular methods for uncertainty estimation in deep learning can be poor choices for sequential decision problems. We show that this disconnect is more than a technical detail, but a serious shortcoming that can lead to arbitrarily poor performance. We support our claims by a series of simple lemmas for simple environments, together with experimental evidence in more complex settings.

Our approach builds on an alternative method for uncertainty in deep RL inspired by the statistical bootstrap [15]. This approach trains an ensemble of models, each on perturbed versions of the data. The resulting distribution *of the ensemble* is used to approximate the uncertainty in the estimate [47]. Although typically regarded as a frequentist method, bootstrapping gives near-optimal convergence rates when used as an approximate Bayesian posterior [19, 18]. However, these ensemble-based approaches to uncertainty quantification approximate a 'posterior' without an effective methodology to inject a 'prior'. This can be a crucial shortcoming in sequential decision problems.

In this paper, we present a simple modification where each member of the ensemble is initialized together with a random but fixed *prior function*. Predictions in each ensemble member are then taken as the sum of the trainable neural network and its prior function. Learning/optimization is performed so that this sum (network plus prior) minimizes training loss. Therefore, with sufficient network capacity and optimization, the ensemble members will agree at observed data. However, in regions of the space with little or no training data, their predictions will be determined by the generalization of their networks and priors. Surprisingly, we show that this approach is equivalent to exact Bayesian inference for the special case of Gaussian linear models. Following on from this 'sanity check', we present a series of simple experiments designed to extend this intuition to deep learning. We show that many of the most popular approaches for uncertainty estimation in deep RL do *not* pass these sanity checks, and crystallize these shortcomings in a series of lemmas and small examples. We demonstrate that our simple modification can facilitate aspiration in difficult tasks where previous approaches for deep RL fail. We believe that this work presents a simple and practical approach to encoding prior knowledge with deep reinforcement learning.

## 2   Why do we need a 'prior' mechanism for deep RL?

We model the environment as a Markov decision process $M = (\mathcal{S}, \mathcal{A}, R, P)$ [10]. Here $\mathcal{S}$ is the state space and $\mathcal{A}$ is the action space. At each time step $t$, the agent observes state $s_t \in \mathcal{S}$, takes action $a_t \in \mathcal{A}$, receives reward $r_t \sim R(s_t, a_t)$ and transitions to $s_{t+1} \sim P(s_t, a_t)$. A policy $\pi : \mathcal{S} \rightarrow \mathcal{A}$ maps states to actions and let $\mathcal{H}_t$ denote the history of observations before time $t$. An RL algorithm maps $\mathcal{H}_t$ to a distribution over policies; we assess its quality through the cumulative reward over unknown environments. To perform well, an RL algorithm must learn to optimize its actions, combining both learning and control [63]. A 'deep' RL algorithm uses neural networks for nonlinear function approximation [32, 42].

The scale and scope of problems that might be approached through deep RL is vast, but there are three key aspects an efficient (and general) agent must address [63]:

1. **Generalization**: be able to learn from data it collects.
2. **Exploration**: prioritize the best experiences to learn from.
3. **Long-term consequences**: consider external effects beyond a single time step.

In this paper we focus on the importance of some form of 'prior' mechanism for efficient exploration. As a motivating example we consider a sparse reward task where random actions are very unlikely to ever see a reward. If an agent has never seen a reward then it is essential that some other form of aspiration, motivation, drive or curiosity direct its learning. We call this type of drive a 'prior' effect, since it does not come from the observed data, but are ambivalent as to whether this effect is philosophically 'Bayesian'. Agents that do not have this prior drive will be left floundering aimlessly and thus may require exponentially large amounts of data in order to learn even simple problems [27].

To solve a specific task, it can be possible to attain superhuman performance without significant prior mechanism [42, 41]. However, if our goal is artificial *general* intelligence, then it is disconcerting that our best agents can perform very poorly even in simple problems [33, 39]. One potentially general approach to decision making is given by the Thompson sampling heuristic[1]: 'randomly take action according to the probability you believe it is the optimal action' [68]. In recent years there have been several attempts to apply this heuristic

to deep reinforcement learning, each attaining significant outperformance over deep RL baselines on certain tasks [20, 47, 35, 11, 17]. In this section we outline crucial shortcomings for the most popular existing approaches to posterior approximation; these outlines will be brief, but more detail can be found in Appendix C. These shortcomings set the scene for Section 3, where we introduce a simple and practical alternative that passes each of our simple sanity checks: bootstrapped ensembles with randomized prior functions. In Section 4 we demonstrate that this approach scales gracefully to complex domains with deep RL.

## 2.1 Dropout as posterior approximation

One of the most popular modern approaches to regularization in deep learning is dropout sampling [61]. During training, dropout applies an independent random Bernoulli mask to the activations and thus guards against excessive co-adaptation of weights. Recent work has sought to understand dropout through a Bayesian lens, highlighting the connection to variational inference and arguing that the resultant dropout distribution approximates a Bayesian posterior [20]. This narrative has proved popular despite the fact that dropout distribution can be a poor approximation to most reasonable Bayesian posteriors [22, 46]:

**Lemma 1** (Dropout distribution does not concentrate with observed data).
*Consider any loss function $\mathcal{L}$, regularizer $\mathcal{R}$ and data $\mathcal{D}=\{(x,y)\}$. Let $\theta$ be parameters of any neural network architecture $f$ trained with dropout rate $p\in(0,1)$ and dropout masks $W$,*

$$\theta_p^* \in \arg\min_{\theta} \mathbb{E}_{W \sim \text{Ber}(p),(x,y) \sim \mathcal{D}} \left[ \mathcal{L}(x, y \mid \theta, W) + \mathcal{R}(\theta) \right]. \tag{1}$$

*Then the dropout distribution $f_{\theta_p^*,W}$ is invariant to duplicates of the dataset $\mathcal{D}$.*

Lemma 1 is somewhat contrived, but highlights a clear shortcoming of dropout as posterior sampling: the dropout rate does not depend on the data. Lemma 1 means no agent employing dropout for posterior approximation can tell the difference between observing a set of data once and observing it $N \gg 1$ times. This can lead to arbitrarily poor decision making, even when combined with an efficient strategy for exploration [45].

## 2.2 Variational inference and Bellman error

Dropout as posterior is motivated by its connection to variational inference (VI) [20], and recent work to address Lemma 1 improves the quality of this variational approximation by tuning the dropout rate from data [21].[2] However, there is a deeper problem to this line of research that is common across many works in this field: even given access to an oracle method for *exact* inference, applying independent inference to the Bellman error does not propagate uncertainty correctly for the value function as a whole [44]. To estimate the uncertainty in $Q$ from the Bellman equation $Q(s_t,a_t)=\mathbb{E}[r_{t+1}+\gamma\max_\alpha Q(s_{t+1},\alpha)]$ it is crucial that the two sides of this equation are not independent random variables. Ignoring this dependence can lead to very bad estimates, even with exact inference.

**Lemma 2** (Independent VI on Bellman error does not propagate uncertainty).
*Let $Y \sim N(\mu_Y,\sigma_Y^2)$ be a target value. If we train $X \sim N(\mu,\sigma^2)$ according to the squared error*

$$\mu^*, \sigma^* \in \arg\min_{\mu,\sigma} \mathbb{E}\left[ (X - Y)^2 \right] \quad \text{for } X, Y \text{ independent}, \tag{2}$$

*then the solution $\mu^* = \mu_Y, \sigma^* = 0$ propagates zero uncertainty from $Y$ to $X$.*

To understand the significance of Lemma 2, imagine a deterministic system that transitions from $s_1$ to $s_2$ without reward. Suppose an agent is able to correctly quantify their posterior uncertainty for the value $V(s_2)=Y \sim N(\mu_Y,\sigma_Y^2)$. Training $V(s_1)=X$ according to (2) will lead to zero uncertainty estimates at $s_1$, when in fact $V(s_1) \sim N(\mu_Y,\sigma_Y^2)$. This observation may appear simplistic, and may not say much more than 'do not use the squared loss' for VI in this setting. However, despite this clear failing (2) is precisely the loss used by the majority of approaches to VI for RL [17, 35, 65, 69, 20]. Note that this failure occurs even without decision making, function approximation and even when the true posterior lies within the variational class.

## 2.3 'Distributional reinforcement learning'

The key ingredient for a Bayesian formulation for sequential decision making is to consider beliefs not simply as a point estimate, but as a *distribution*. Recently an approach called 'distributional RL' has shown great success in improving stability and performance in deep RL benchmark algorithms [8]. Despite the name, these two ideas are quite distinct. 'Distributional RL' replaces a scalar estimate for the value function by a distribution that is trained to minimize a loss against the distribution of data it observes. This distribution of observed data is an orthogonal concept to that of Bayesian uncertainty.

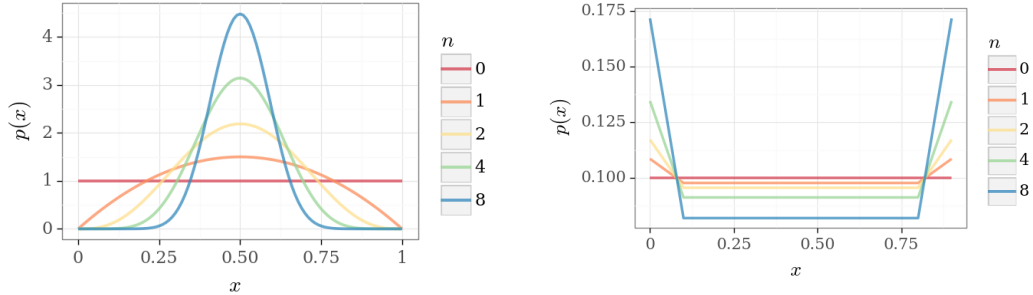

(a) Posterior beliefs concentrate around $p = 0.5$.     (b) 'Distributional' tends to mass at 0 and 1.

Figure 1: Output distribution after observing $n$ heads and $n$ tails of a coin.

Figure 1 presents an illustration of these two distributions after observing flips of a coin. As more data is gathered the posterior distribution concentrates around the mean whereas the 'distributional RL' estimate approaches that of the generating Bernoulli. Although both approaches might reasonably claim a 'distributional perspective' on RL, these two distributions have orthogonal natures and behave quite differently. Conflating one for the other can lead to arbitrarily poor decisions; it is the uncertainty in beliefs ('epistemic'), not the distributional noise ('aleatoric') that is important for exploration [27].

## 2.4 Count-based uncertainty estimates

Another popular method for incentivizing exploration is with a density model or 'pseudocount' [6]. Inspired by the analysis of tabular systems, these models assign a bonus to states and actions that have been visited infrequently according to a density model. This method can perform well, but only when the generalization of the density model is aligned with the task objective. Crucially, this generalization is not learned from the task [53].

Even with an optimal state representation and density, a count-based bonus on states can be poorly suited for efficient exploration. Consider a linear bandit with reward $r_t(x_t) = x_t^T \theta^* + \epsilon_t$ for some $\theta^* \in \mathbb{R}^d$ and $\epsilon_t \sim N(0, 1)$ [56]. Figure 2 compares the uncertainty in the expected reward $\mathbb{E}[x^T \theta^*]$ with that obtained by density estimation on the observed $x_t$. A bonus based upon the state density does not correlate with the *uncertainty* over the unknown optimal action. This disconnect can lead to arbitrarily poor decisions [49].

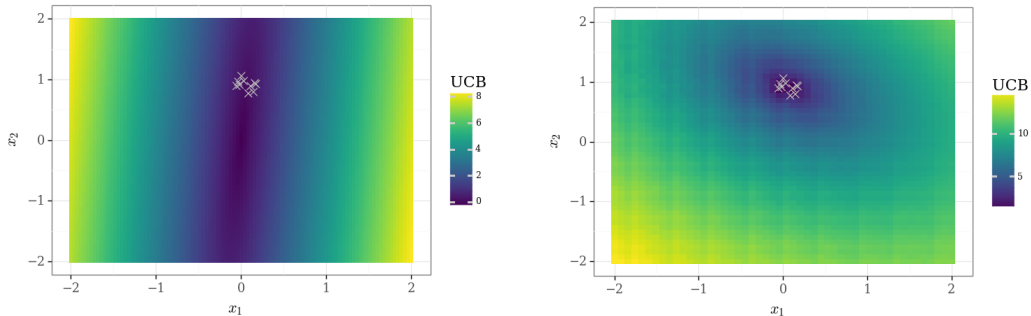

(a) Uncertainty bonus from posterior over $x^T \theta^*$.     (b) Bonus from Gaussian pseudocount $p(x)$.

Figure 2: Count-based uncertainty leads to a poorly aligned bonus even in a linear system.

# 3 Randomized prior functions for deep ensembles

Section 2 motivates the need for effective uncertainty estimates in deep RL. We note that crucial failure cases of several popular approaches can arise even with simple linear models. As a result, we take a moment to review the setting of Bayesian linear regression. Let $\theta \in \mathbb{R}^d$ with prior $N(\overline{\theta}, \lambda I)$ and data $\mathcal{D} = \{(x_i, y_i)\}_{i=1}^n$ for $x_i \in \mathbb{R}^d$ and $y_i = \theta^T x_i + \epsilon_i$ with $\epsilon_i \sim N(0, \sigma^2)$ iid. Then, conditioned on $\mathcal{D}$, the posterior for $\theta$ is Gaussian:

$$\mathbb{E}[\theta|\mathcal{D}] = \left(\frac{1}{\sigma^2}X^T X + \frac{1}{\lambda}I\right)^{-1}\left(\frac{1}{\sigma^2}X^T y + \frac{1}{\lambda}\overline{\theta}\right), \quad \text{Cov}[\theta|\mathcal{D}] = \left(\frac{1}{\sigma^2}X^T X + \frac{1}{\lambda}I\right)^{-1}. \quad (3)$$

Equation (3) relies on Gaussian conjugacy and linear models, which cannot easily be extended to deep neural networks. The following result shows that we can replace this analytical result with a simple computational procedure.

**Lemma 3** (Computational generation of posterior samples).
*Let $f_\theta(x) = x^T\theta$, $\tilde{y}_i \sim N(y_i, \sigma^2)$ and $\tilde{\theta} \sim N(\overline{\theta}, \lambda I)$. Then the either of the following optimization problems generate a sample $\theta \mid \mathcal{D}$ according to (3):*

$$\arg\min_\theta \sum_{i=1}^n \|\tilde{y}_i - f_\theta(x_i)\|_2^2 + \frac{\sigma^2}{\lambda}\|\tilde{\theta} - \theta\|_2^2, \quad (4)$$

$$\tilde{\theta} + \arg\min_\theta \sum_{i=1}^n \|\tilde{y}_i - (f_{\tilde{\theta}} + f_\theta)(x_i)\|_2^2 + \frac{\sigma^2}{\lambda}\|\theta\|_2^2. \quad (5)$$

*Proof.* To prove (4) note that the solution is Gaussian and then match moments; equation (5) then follows by relabeling [49]. $\square$

Lemma 3 is revealing since it allows us to view Bayesian regression through a purely computational lens: 'generate posterior samples by training on noisy versions of the data, together with some random regularization'. Even for nonlinear $f_\theta$, we can still compute (4) or (5). Although the resultant $f_\theta$ will no longer be an exact posterior, at least it passes the 'sanity check' in this simple linear setting (unlike the approaches of Section 2). We argue this method is quite intuitive: the perturbed data $\tilde{\mathcal{D}} = \{(x_i, \tilde{y}_i)\}_{i=1}^n$ is generated according to the estimated noise process $\epsilon_t$ and the sample $\tilde{\theta}$ is drawn from prior beliefs. Intuitively (4) says to fit to $\tilde{\mathcal{D}}$ and regularize weights to a prior sample of weights $\tilde{\theta}$; (5) says to generate a prior *function* $f_{\tilde{\theta}}$ and then fit an additive term to noisy data $\tilde{\mathcal{D}}$ with regularized complexity.

This paper explores the performance of each of these methods for uncertainty estimation with deep learning. We find empirical support that method (5) coupled with a *randomized prior function* significantly outperforms ensemble-based approaches without prior mechanism. We also find that (5) significantly outperforms (4) in deep RL. We suggest a major factor in this comes down to the huge dimensionality of neural network weights, whereas the output policy or value is typically far smaller. In this case, it makes sense to enforce prior beliefs in the low dimensional space. Further, the initialization of neural network weights plays an important role in their generalization properties and optimization via stochastic gradient descent (SGD) [23, 38]. As such, (5) may help to decouple the dual roles of initial weights as both 'prior' and training initializer. Algorithm 1 describes our approach applied to modern deep learning architectures.

---

**Algorithm 1** Randomized prior functions for ensemble posterior.

---

**Require:** Data $\mathcal{D} \subseteq \{(x,y)|x \in \mathcal{X}, y \in \mathcal{Y}\}$, loss function $\mathcal{L}$, neural model $f_\theta : \mathcal{X} \to \mathcal{Y}$,
Ensemble size $K \in \mathbb{N}$, noise procedure `data_noise`, distribution over priors $\mathcal{P} \subseteq \{\mathbb{P}(p)|p : \mathcal{X} \to \mathcal{Y}\}$.
1: **for** $k = 1, .., K$ **do**
2:     initialize $\theta_k \sim$ Glorot initialization [23].
3:     form $\mathcal{D}_k = \texttt{data\_noise}(\mathcal{D})$ (e.g. Gaussian noise or bootstrap sampling [50]).
4:     sample prior function $p_k \sim \mathcal{P}$.
5:     optimize $\nabla_{\theta|\theta=\theta_k}\mathcal{L}(f_\theta + p_k; \mathcal{D}_k)$ via ADAM [28].
6: **return** ensemble $\{f_{\theta_k} + p_k\}_{k=1}^K$.

---

## 4 Deep reinforcement learning

Algorithm 1 might be applied to model or policy learning approaches, but this paper focuses on value learning. We apply Algorithm 1 to *deep Q networks* (DQN) [42] on a series of tasks designed to require good uncertainty estimates. We train an ensemble of $K$ networks $\{Q_k\}_{k=1}^{K}$ in parallel, each on a perturbed version of the observed data $\mathcal{H}_t$ and each with a distinct random, but fixed, prior function $p_k$. Each episode, the agent selects $j \sim \text{Unif}([1,..,K])$ and follows the greedy policy w.r.t. $Q_j$ for the duration of the episode. This algorithm is essentially bootstrapped DQN (BootDQN) except for the addition of the prior function $p_k$ [47]. We use the statistical bootstrap rather than Gaussian noise (5) to implement a state-specific variance [19].

Let $\gamma \in [0,1]$ be a discount factor that induces a time preference over future rewards. For a neural network family $f_\theta$, prior function $p$, and data $\mathcal{D} = \{(s_t, a_t, r_t, s_t')\}$ we define the $\gamma$-discounted empirical temporal difference (TD) loss,

$$\mathcal{L}_\gamma(\theta; \theta^-, p, \mathcal{D}) := \sum_{t \in \mathcal{D}} \left( r_t + \gamma \max_{a' \in \mathcal{A}} \overbrace{(f_{\theta^-} + p)}^{\text{target } Q}(s_t', a') - \overbrace{(f_\theta + p)}^{\text{online } Q}(s_t, a_t) \right)^2 . \tag{6}$$

Using this notation, the learning update for BootDQN with prior functions is a simple application of Algorithm 1, which we outline below. To complete the RL algorithm we implement a 50-50 `ensemble_buffer`, where each transition has a 50% chance of being included in the replay for model $k = 1,..,K$. For a complete description of BootDQN+prior agent, see Appendix A.

---

**Algorithm 2** `learn_bootstrapped_dqn_with_prior`

| **Agent:** | $\theta_1,..,\theta_K$ | trainable network parameters |
|---|---|---|
| | $p_1,..,p_K$ | fixed prior functions |
| | $\mathcal{L}_\gamma(\theta{=}\cdot\,;\theta^-{=}\cdot\,,p{=}\cdot\,,\mathcal{D}{=}\cdot)$ | TD error loss function |
| | `ensemble_buffer` | replay buffer of $K$-parallel perturbed data |
| **Updates:** | $\theta_1,..,\theta_K$ | agent value function estimate |

1: **for** $k$ in $(1,\ldots,K)$ **do**
2:     Data $\mathcal{D}_k \leftarrow$ `ensemble_buffer[k].sample_minibatch()`
3:     optimize $\nabla_{\theta|\theta=\theta_k} \mathcal{L}(\theta; \theta_k, p_k, \mathcal{D}_k)$ via ADAM [28].

---

### 4.1 Does BootDQN+prior address the shortcomings from Section 2?

Algorithm 2 is a simple modification of vanilla Q-learning: rather than maintain a single point estimate for $Q$, we maintain $K$ estimates in parallel, and rather than regularize each estimate to a single value, each is individually regularized to a distinct random prior function. We show that that this simple and scalable algorithm overcomes the crucial shortcomings that afflict existing methods, as outlined in Section 2.

✓ **Posterior concentration** (Section 2.1): Prior function + noisy data means the ensemble is initially diverse, but concentrates as more data is gathered. For linear-gaussian systems this matches Bayes posterior, bootstrap offers a general, non-parametric approach [16, 18].

✓ **Multi-step uncertainty** (Section 2.2): Since each network $k$ trains only on its *own* target value, BootDQN+prior propagates a temporally-consistent sample of $Q$-value [49].

✓ **Epistemic vs aleatoric** (Section 2.3): BootDQN+prior optimises the *mean* TD loss (6) and does not seek to fit the noise in returns, unlike 'distributional RL' [7].

✓ **Task-appropriate generalization** (Section 2.4): We explore according to our uncertainty in the value $Q$, rather than density on state. As such, our generalization naturally occurs in the space of *features* relevant to the task, rather than pixels or noise [6].

✓ **Intrinsic motivation** (comparison to BootDQN without prior): In an environment with zero rewards, a bootstrap ensemble may simply learn to predict zero for *all* states. The prior $p_k$ can make this generalization unlikely for $Q_k$ at unseen states $\tilde{s}$ so $\mathbb{E}[\max_\alpha Q_k(\tilde{s},\alpha)]{>}0$; thus BootDQN+prior seeks novelty even with no observed rewards.

Another source of justification comes from the observation that BootDQN+prior is an instance of *randomized least-squares value iteration* (RLSVI), with regularization via 'prior

function' for an ensemble of neural networks. RLSVI with linear function approximation and Gaussian noise guarantees a bound on expected regret of $\tilde{O}(\sqrt{|\mathcal{S}||\mathcal{A}|T})$ in the tabular setting [49].[3] Similarly, analysis for the bandit setting establishes that $K = \tilde{O}(|\mathcal{A}|)$ models trained online can attain similar performance to full resampling each episode [36]. Our work in this paper pushes beyond the boundaries of these analyses, which are presented as 'sanity checks' that our algorithm is at least sensible in simple settings, rather than a certificate of correctness for more complex ones. The rest of this paper is dedicated to an empirical investigation of our algorithm through computational experiments. Encouragingly, we find that many of the insights born out of simple problems extend to more complex 'deep RL' settings and good evidence for the efficacy of our algorithm.

## 4.2 Computational experiments

Our experiments focus on a series of environments that require deep exploration together with increasing state complexity [27, 49]. In each of our domains, random actions are very unlikely to achieve a reward and exploratory actions may even come at a cost. Any algorithm without prior motivation will have no option but to explore randomly, or worse, eschew exploratory actions completely in the name of premature and sub-optimal exploitation. In our experiments we focus on a *tabula rasa* setting in which the prior function is drawn as a random neural network. Although our prior distribution $\mathcal{P}$ could encode task-specific knowledge (e.g. through sampling the true $Q^*$), we leave this investigation to future work.

### 4.2.1 Chain environments

We begin our experiments with a family of chain-like environments that highlight the need for deep exploration [62]. The environments are indexed by problem scale $N \in \mathbb{N}$ and action mask $W \sim \text{Ber}(0.5)^{N \times N}$, with $\mathcal{S} = \{0,1\}^{N \times N}$ and $\mathcal{A} = \{0,1\}$. The agent begins each episode in the upper left-most state in the grid and deterministically falls one row per time step. The state encodes the agent's row and column as a one-hot vector $s_t \in \mathcal{S}$. The actions $\{0,1\}$ move the agent left or right depending on the action mask $W$ at state $s_t$, which remains fixed. The agent incurs a cost of $0.01/N$ for moving right in all states except for the right-most, in which the reward is 1. The reward for action left is always zero. An episode ends after $N$ time steps so that the optimal policy is to move right each step and receive a total return of 0.99; all other policies receive zero or negative return. Crucially, algorithms without deep exploration take $\Omega(2^N)$ episodes to learn the optimal policy [52].[4]

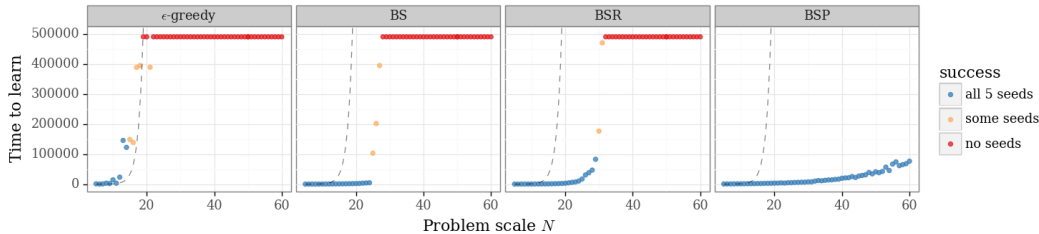

Figure 3: Only bootstrap with additive prior network (BSP) scales gracefully to large problems. Plotting BSP on a log-log scale suggests an empirical scaling $T_{\text{learn}} = \tilde{O}(N^3)$; see Figure 8.

Figure 3 presents the average time to learn for $N = 5, .., 60$ up to 500K episodes over 5 seeds and ensemble $K = 20$. We say that an agent has learned the optimal policy when the average regret per episode drops below 0.9. We compare three variants of BootDQN, depending on their mechanism for 'prior' effects. **BS** is bootstrap without prior mechanism. **BSR** is bootstrap with $l_2$ regularization on weights per (4). **BSP** is bootstrap with additive prior function per (5). In each case we initialize a random 20-unit MLP; BSR regularizes to these initial weights and BSP trains an additive network. Although all bootstrap methods significantly outperform $\epsilon$-greedy, only BSP successfully scales to large problem sizes.

Figure 4 presents a more detailed analysis of the sensitivity of our approach to the tuning parameters of different regularization approaches. We repeat the experiments of Figure 3

and examine the size of the largest problem solved before 50K episodes. In each case larger ensembles lead to better performance, but this effect tends to plateau relatively early. Figure 4a shows that regularization provides little or no benefit to BSR. Figure 4b examines the effect of scaling the randomly initialized MLP by a scalar hyperparameter $\beta$.

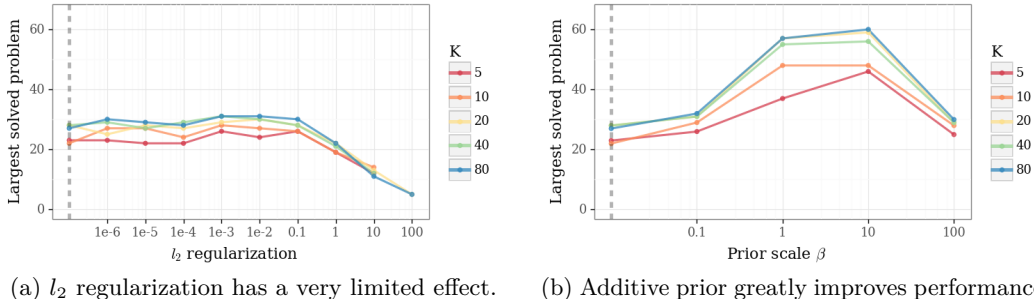

(a) $l_2$ regularization has a very limited effect.     (b) Additive prior greatly improves performance.

Figure 4: Comparing effects of different styles of prior regularization in Bootstrapped DQN.

**How does BSP solve this exponentially-difficult problem?**

At first glance this 'chain' problem may seem like an impossible task. Finding the single rewarding policy out of $2^N$ is not simply a needle-in-a-haystack, but more akin to looking for a piece of hay in a needle-stack! Since every policy apart from the rewarding one is painful, it's very tempting for an agent to give up and receive reward zero. We now provide some intuition for how BSP is able to consistently and scalably solve such a difficult task.

One way to interpret this result is through analysing BSP with linear function approximation via Lemma 3. As outlined in Section 4.1, BSP with linear function approximation satisfies a polynomial regret bound [49]. Further, this empirical scaling matches that predicted by the regret bound tabular domain [51] (see Figure 8). Here, the prior function plays a crucial role - it provides motivation for the agent to explore even when the observed data has low (or no) reward. Note that it is not necessary the sampled prior function leads to a good policy itself; in fact this is exponentially unlikely according to our initialization scheme. The crucial element is that when a new state $s'$ is reached there is *some* ensemble member that estimates $\max_{a'} Q_k(s', a')$ is sufficiently positive to warrant visiting, even if it causes some negative reward along the way. In that case, when network $k$ is active it will seek out the potentially-informative $s'$ even if it is multiple timesteps away; this effect is sometimes called *deep exploration*. We present an accompanying visualization at http://bit.ly/rpf_nips.

However, this connection to linear RLSVI does not inform why BSP should outperform BSR. To account for this, we appeal to the functional dynamics of deep learning architectures (see Section 3). In large networks weight decay (per BSR) may be an ineffective mechanism on the *output* $Q$-values. Instead, training an additive network via SGD (per BSP) may provide a more effective regularization on the output function [73, 38, 5]. We expand on this hypothesis and further details of these experiments in Appendix B.1. This includes investigation of NoisyNets [17] and dropout [20], which both perform poorly, and a comparison to UCB-based algorithms, which scale much worse than BSP, even with oracle access to state visit counts.

### 4.2.2 Cartpole swing-up

The experiments of Section 4.2.1 show that the choice of prior mechanism can be absolutely essential for efficient exploration via randomized value functions. However, since the underlying system is a small finite MDP we might observe similar performance through a tabular algorithm. In this section we investigate a classic benchmark problem that necessitates nonlinear function approximation: cartpole [63]. We modify the classic formulation so that the pole begins hanging down and the agent only receives a reward when the pole is upright, balanced, and centered[5]. We also add a cost of 0.1 for moving the cart. This problem embodies many of the same aspects of 4.2.1, but since the agent interacts with the environment through state $s_t = (\cos(\theta_t), \sin(\theta_t), \dot{\theta}_t, x_t, \dot{x}_t)$, the agent must also learn nonlinear generalization. Tabular approaches are not practical due to the curse of dimensionality.

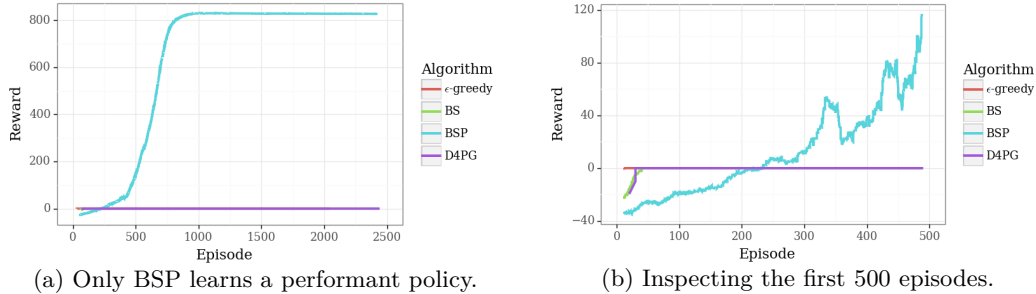

(a) Only BSP learns a performant policy.    (b) Inspecting the first 500 episodes.

Figure 5: Learning curves for the modified cartpole swing-up task.

Figure 5 compares the performance of DQN with $\epsilon$-greedy, bootstrap without prior (BS), bootstrap with prior networks (BSP) and the state-of-the-art continuous control algorithm D4PG, itself an application of 'distributional RL' [4]. Only BSP learns a performant policy; no other approach ever attains any positive reward. We push experimental details, including hyperparameter analysis, to Appendix B.2. These results are significant in that they show that our intuitions translate from simple domains to more complex nonlinear settings, although the underlying state is relatively low dimensional. Our next experiments investigate performance in a high dimensional and visually rich domain.

### 4.2.3 Montezuma's revenge

Our final experiment comes from the Arcade Learning Environment and the canonical sparse reward game, Montezuma's Revenge [9]. The agent interacts directly with the pixel values and, even under an optimal policy, there can be hundreds of time steps between rewarding actions. This problem presents a significant exploration challenge in a visually rich environment; many published algorithms are essentially unable to attain any reward here [42, 41]. We compare performance against a baseline distributed DQN agent with double Q-learning, prioritized experience replay and dueling networks [25, 24, 59, 72]. To save computation we follow previous work and use a shared convnet for the ensemble uncertainty [47, 3]. Figure 6 presents the results for varying prior scale $\beta$ averaged over three seeds. Once again, we see that the prior network can be absolutely critical to successful exploration.

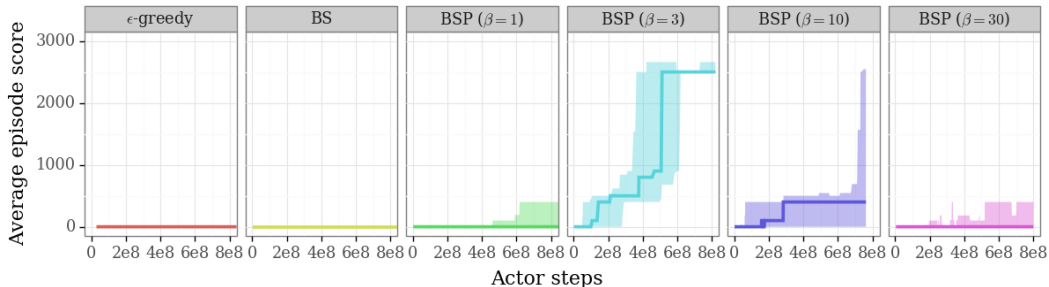

Figure 6: The prior network qualitatively changes behavior on Montezuma's revenge.

## 5 Conclusion

This paper highlights the importance of uncertainty estimates in deep RL, the need for an effective 'prior' mechanism, and its potential benefits towards efficient exploration. We present some alarming shortcomings of existing methods and suggest bootstrapped ensembles with randomized prior functions as a simple, practical alternative. We support our claims through an analysis of this method in the linear setting, together with a series of simple experiments designed to highlight the key issues. Our work leaves several open questions. What kinds of prior functions are appropriate for deep RL? Can they be optimized or 'meta-learned'? Can we distill the ensemble process to a single network? We hope this work helps to inspire solutions to these problems, and also build connections between the theory of efficient learning and practical algorithms for deep reinforcement learning.

**Acknowledgements**

We would like to thank many people who made important contributions to this paper. This paper can be thought of as a specific type of 'deep exploration via randomized value functions', whose line of research has been crucially driven by the contributions of (and conversations with) Benjamin Van Roy, Daniel Russo and Zheng Wen. Further, we would like to acknowledge the many helpful comments and support from Mohammad Gheshlaghi Azar, David Budden, David Silver and Justin Sirignano. Finally, we would like to make a special mention for Hado Van Hasselt, who coined the term 'hay in a needle-stack' to describe our experiments from Section 4.2.1.

## Footnotes

[1]This heuristic is general in the sense that Thompson sampling can be theoretically justified in many of the domains where these other approaches fail [1, 48, 34, 58].

[2]Concrete dropout assymptotically improves the quality of the variational approximation, but provides no guarantees on its rate of convergence or error relative to exact Bayesian inference [21].

[3]Regret measures the shortfall in cumulative rewards compared to that of the optimal policy.

[4]The dashed lines indicate the $2^N$ dithering lower bound. The action mask $W$ means this cannot be solved easily by evolution or policy search evolution, unlike previous 'chain' examples [47, 54].

[5]We use the DeepMind control suite [66] with reward +1 only when $\cos(\theta) > 0.95$, $|x| < 0.1$, $|\dot{\theta}| < 1$, and $|\dot{x}| < 1$. Each episode lasts 1,000 time steps, simulating 10 seconds of interaction.

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
