[Supplementary Material]

# A  Reinforcement learning algorithm

In this appendix we fill out the details for the complete pseudocode for the BootDQN+Prior RL agent. Our problem setting matches the description of prior work [49]; we reproduce the algorithms and figures in this section for the convenience of our readers. At a high level, our agents interact with the environment through repeated finite episodes as described by Algorithm 3. To describe an agent, we must simply implement the `act`, `update_buffer` and `learn_from_buffer` methods.

---

**Algorithm 3** `live`

---

**Input:**    `agent`          methods `act`, `update_buffer`, `learn_from_buffer`
             `environment`  methods `reset`, `step`

1: **for** episode $= 1, 2, \ldots$ **do**
2:     `agent.learn_from_buffer()`
3:     `transition` $\leftarrow$ `environment.reset()`
4:     **while** `transition.new_state` is not **null do**
5:         `action` $\leftarrow$ `agent.act(transition.new_state)`
6:         `transition` $\leftarrow$ `environment.step(action)`
7:         `agent.update_buffer(transition)`

---

BootDQN+prior implements an `ensemble_buffer` that maintains $K$ buffers in parallel, although this may clearly be implemented in an efficient way that uses $o(K)$ memory. Figure 7 provides an illustration of how BootDQN learns and maintains $K$ estimates of the value function in parallel.

(a) Learning a single value function      (b) Learning multiple value functions in parallel

Figure 7: RLSVI via ensemble sampling, each member produced by LSVI on perturbed data.

To implement an online double-or-nothing bootstrap we employ Algorithm 4, which assigns each transition to each ensemble buffer with probability $\frac{1}{2}$.

---

**Algorithm 4** `ensemble_buffer.update_bootstrap`$(\cdot)$

---

**Input:**      `transition`        $(s_t, a_t, r_t, s'_t, t)$
**Updates:**  `ensemble_buffer`  replay buffer of $K$-parallel perturbed data

1: **for** $k$ in $(1, \ldots, K)$ **do**
2:     **if** $m_t^k \sim \mathrm{Unif}(\{0, 1\}) = 1$ **then**
3:         `ensemble_buffer[k].enqueue(`$(s_t, a_t, r_t, s'_t, t)$`)`

---

Algorithm 2 describes the `learn_from_buffer` method for the agent. For our experiments, we sometimes amend Algorithm 3 to learn periodically every $N$ steps, rather than only at the end of the episode, but we mention this in the text where this is the case. This practice is common for most implementations of DQN and other reinforcement learning algorithms, but it does not play a significant role in our algorithm.

The final piece to describe BootDQN+prior is the `act` method for action selection. We employ a form of approximate Thompson sampling for RL via randomized value functions. Every episode, the agent selects $j \sim \mathrm{Unif}(1, .., K)$ and follows the greedy policy for $Q_j$ for the duration of the episode.

# B Reinforcement learning experiments

In this section, we expand on details for the experimental set-up together with some additional results. Unless otherwise stated we use TensorFlow defaults, Adam optimizer with learning rate $10^{-3}$ and uniform experience replay with batch size 128. For our $\epsilon$-greedy DQN baseline, we anneal epsilon linearly over 2000 episodes and perform hyperparameter sweeps over the initial epsilon $\epsilon_0$. All other agents (NoisyNet, Dropout, Ensemble, Bootstrap) use greedy policies according to an appropriate per-episode Thompson sampling.

## B.1 Chain environments

Figure 3 shows the time it takes each agent to learn a problem of size $N$. Figure 8 reproduces these results but on a log-log scale, which helps to reveal the problem scaling as $N$ increases. As in Figure 3, the dashed line corresponds to a dithering lower bound $T_{\text{learn}} = 2^N$. We also include a solid line with slope equal to three, corresponding to a polynomial growth $T_{\text{learn}} = \tilde{O}(N^3)$.

Figure 8: Log-log plot demonstrates scaling of learning behaviour.

In addition to BSP, BSR, BS and $\epsilon$-greedy displayed in Figure 3, we also ran parameter sweeps for dropout, NoisyNet and a count-based exploration strategy. Figure 9 presents the result for NoisyNet and dropout, each individually tuned up to 50k episodes. Even after tuning dropout rate and sampling frequency (by episode or by timestep) neither dropout nor NoisyNet scale successfuly to large domains.

Figure 9: Learning time for noisy and dropout; neither approach scales well.

To compare with 'count-based' exploration we implement a version of DQN that optimizes the true reward plus a UCB exploration bonus $\frac{\beta}{\sqrt{N_t(s)}}$, where $N(s)$ is the number of visits to state $s$ prior to time $t$ [26, 6]. Figure 10 shows that this count-based exploration strategy performs much worse than BSP, even after sweeping over bonus scale $\beta$ and even with access to the true state visit-counts. This mirrors the outperformance of PSRL vs UCRL in tabular reinfocement learning. One explanation for this discrepancy comes from the inefficient way UCB-style algorithms propagate uncertainty over many timesteps [51, 44].

Figure 10: Sweeping over optimistic bonus; no scale of $\beta$ matches BSP performance.

For all of our algorithms we tune agent hyperparameters by grid search. These were:

- **$\epsilon$-greedy**: $\epsilon = 0.1$, linearly annealed to zero.
- **BSP**: prior scale $\beta = 10$ (Figure 4b).
- **BSR**: $l_2$ regularizer scale $\lambda = 0.1$ (Figure 4a).
- **Dropout**: Resample mask every step with $p_{\text{keep}} = 0.1$.
- **NoisyNet**: Resample noise every step.
- **UCB**: Optimistic bonus $\beta = 0.1$ (Figure 10).

## B.2  Sparse cartpole swing-up

In Section 4.2.2 we presented experiments showing that BSP outperforms benchmark algorithms. Figure 11 presents the sensitivity of BSP sensitivity to the prior scale $\beta$ on this domain. Small values of $\beta$ prematurely and suboptimally converge to the stationary policy, and so receive zero cumulative reward. Larger values of $\beta$ take longer to wash away their prior effect, but we expect them to learn a performant policy eventually. This behaviour mirrors the scaling we saw in the chain environments, which is reassuring.

Figure 11: Sensitivity of performance to prior scale $\beta$.

## B.3  Montezuma's revenge

In our experiments, we use the standard Atari configuration and preprocessing including greyscaling, frame stacking, action repeats, and random no-op starts [42], and the same agent hyperparameters as those used in the Ape-X paper [25]. However, our agent implementation is somewhat different and so our baseline results are not directly comparable across all games.

# C   Why do we need a 'prior' mechanism for deep RL?

Section 2 outlines the need for a prior mechanism in deep RL, together with key failure cases for several of the most popular approaches. Due to space limitations we provide only simple illustrations of potential inadequacies of each method and this does not preclude their efficacy on any particular domain. In this appendix we expand on the details provided in Section 2 and provide suggestions for how these approaches might be remedied by future work.

## C.1   Dropout as posterior approximation

Previous work has suggested that dropout works as an effective variational approximation to the Bayesian posterior in neural networks, without special consideration for the network architecture [20]. However, Lemma 1 is a general statement that gives us cause to question the quality of this approximation. In this subsection we dig deeper into an extremely simple estimation problem, a linear network with $d$ units to estimate the mean of a random variable $Y \in \mathbb{R}$. Even in this simple setting dropout performs poorly as a Bayesian posterior.

We form predictions $f_\theta = \sum_{i=1}^d w_i \theta_i$ with $w_i \sim \text{Ber}(p)$, square loss and regularizer $\mathcal{R}(\theta) = \lambda \|\theta\|^2$ for $\lambda > 0$. Then for any data $\mathcal{D}$ with empirical mean $\overline{y}$, the expected loss solution to (1) is given by [61] $\theta_p^* = \overline{\theta} \mathbb{1}$ for $\overline{\theta} = \frac{\overline{y}}{1+p(d-1)+\frac{\lambda}{d}}$. [6] The resultant predictive distribution therefore has mean $\mu = \overline{\theta} dp$ and standard deviation $\sigma = \overline{\theta}\sqrt{dp(1-p)}$.

If we are to understand dropout as an approximation to a Bayesian posterior, then we should note that this behavior is unusual. First, the only connection to the data is through the empirical mean $\overline{y}$; any possible dataset with the same mean would result in the same 'posterior' distribution. Second we note that $\sigma = \mu\sqrt{(1-p)/dp}$. This coupling means it is not possible for $\sigma \to 0$ and $\mu \nrightarrow 0$, regardless of $\lambda$. More typically we would imagine $\mu \to \mathbb{E}[Y]$ and $\sigma \to 0$ according to the Bayesian central limit theorem [12].

This disconnect is not simply an analytical mistake, but can lead to arbitrarily bad decisions in even the simplest problems. Imagine a simple two-armed bandit problem with one arm's rewards $\sim \text{Ber}(1/2)$ and the other's $\sim \text{Ber}(1/2+\epsilon)$, and the agent does not know which arm is which a priori. This style of problem is particularly well understood with guarantees that Thompson sampling with more reasonable forms of posterior approximation incur regret $\tilde{O}(\log(T))$ in this setting [2]. We refer to this problem as †. The following result highlights that dropout as posterior approximation can perform poorly even on this simple domain.

**Lemma 4** (Dropout sampling attains linear regret in †).
*Fix any $d$, $p$, $\lambda$ and consider the problem of † with an agent employing Thompson sampling by dropout for action selection. Then the expected regret is $\Omega(T)$.*

*Proof.* For any $d$, $p$, $\lambda$ and any observed data $\mathcal{H}_t$, there exists a non-zero probability $P_1(s,p,\lambda,\mathcal{H}_t) > \frac{p^d}{2}$ of selecting action 1 over action action 2. We can see this by imagining all units estimating action 2 are set to zero, then there is at least 50% chance of selection action 1. This proves[7] that $\mathbb{E}[\text{Regret}(T)] \geq \frac{\epsilon p^d}{2} T$ for all $T$. $\square$

Although our analysis of dropout has focused on an exceedingly simple functional form, the key insight that the degree of variability in the posterior distribution does not concentrate with data extends to any neural network architecture. Figure 12 presents the dropout posterior on a simple regression task with a (20,20)-MLP with rectified linear units. We display the predictive distribution under varying amounts of data. The dropout sampling distribution does not converge with increasing amounts of data, whereas the bootstrapped sampling approach behaves much more reasonably. This leads to poor performance in reinforcement learning tasks too, as we saw in Appendix B.

Figure 12: Dropout does not converge with increasing data even with a complex neural network. Grey regions indicate ±1,2 standard deviations, the mean is shown in blue and a single posterior sample in red.

## C.2 Variational inference and Bellman error

Lemma 2 highlights that the basic loss most commonly used by variational approximations to the value function are fundamentally ill-suited to the problem at hand [35, 17]. In Appendix B we present results of such a variational approach, NoisyNet, to some of our benchmark reinforcement learning tasks. As expected, the algorithm performs poorly even after extensive tuning. At the heart of this issue is a sample-based loss that trains to match the *expectation* of the target distribution, but does not attempt to match the higher moments of the uncertainty. However, we could imagine an alternative approach that does aim to match the entire resultant distribution, for example via parameterized distribution and cross entropy loss; we leave this to future work.

Even where variational inference is employed correctly over mutliple timesteps, it may be difficult to encode useful prior knowledge in common variational methods. First, many applications of variational inference (VI) model the distribution over network weights as a product of independent Gaussians [11]. These models facilitate efficient computation, but can underestimate the uncertainty and may be a poor choice for encoding prior knowledge. Even if one were given a mapping of prior knowledge to weights, the confounding demands of good initialization for SGD training may interfere negatively [23]. For this reason practical applications of VI to RL typically use very little prior effect, or even no prior regularization at all [35, 17].

## C.3 'Distributional reinforcement learning'

Unlike the objections of Appendix C.2, 'distributional RL'[8] does learn a value function estimate through a distributional loss. However, this distribution is a distribution over *outcomes* and not a distribution over the *epistemic uncertainty* in the mean beliefs. This distinction between two types of uncertainty, (1.) things that you don't know and (2.) things that are stochastic, is a delicate one and is important to characterize correctly. Both are discussed under many names:

(1.) 'Reducible uncertainty' $\iff$ 'epistemic uncertainty' $\iff$ 'uncertainty',
(2.) 'Irreducible uncertainty' $\iff$ 'aleatoric uncertainty' $\iff$ 'risk'.

Typical decision problems may include elements of both types of uncertainty. Flipping a coin we might want to know both (1.) our posterior beliefs over the probability of heads and (2.) a distribution that categorizes the likely possible outcomes. However, it should be clear that the two concepts are fundamentally distinct. For the purposes of exploration, the Bayesian uncertainty over (1.) should prioritize the acquisition of new knowledge. 'Distributional RL' approximates (2.) and its role is not exchangeable with (1.).

**Lemma 5** (Using 'distributional RL' as a posterior can lead to arbitrarily bad decisions)**.**
*Consider an agent with full information that decides between action 1 with reward $\sim \text{Ber}(0.5)$ and action 2 with reward $1-\epsilon$ for $0<\epsilon<\frac{1}{2}$. If the agent employs Thompson sampling correctly then it will pick action 2 at every step with zero regret. If the agent mistakenly employs Thompson sampling over its 'distributional value function' then it will incur*

$$\mathbb{E}[\text{Regret}(T)] \geq \frac{1}{2}\left(\frac{1}{2}-\epsilon\right)T.$$

Lemma 5 shows that using the 'distributional' value function approximating (2.) can be a poor proxy for the Bayesian uncertainty. However, the Bayesian uncertainty can be a similarly poor proxy for the 'distributional' value function. This can be equally damaging, particularly if the agent has some some risk-sensitive utility with respect to cumulative rewards. It is entirely possible to combine both notions of uncertainty in an agent, although for the goal of maximizing expected cumulative it is not entirely clear what is the benefit of modeling (2.). Certainly, 'distributional' agents have recently attained strong scores in Atari 2600 benchmarks but it is so far unclear exactly what the source of this outperformance comes from [8, 13]. Possible explanations may include more stable gradients, bounded values and the 'many predictions' hypothesis [64]: that learning a distribution may effectively create a series of auxiliary losses. We leave these questions for future work.

### C.4   Count-based uncertainty estimates

Count-based approaches to exploration give a bonus to states that have not been visited frequently according to some density measure $p(x)$. These methods have performed well in many sparse reward tasks such as Montezuma's Revenge, where visiting new states acts as a shaping reward for the true reward [6]. However, a count-based bonus is generally a poor approach to exploration beyond the tabular setting. To see why this is the case note that the density measure of the states may not correlate well with the agent's uncertainty over the optimal policy in that state. We can imagine situations both where the state is visually new, but an agent should still know exactly what to do; and also settings that are only delicately different to a common situation but still necessitate exploration of the optimal policy.

This disconnect shows up in problems as simple as the linear bandit.[9] Via a packing argument, an agent with count-based uncertainty will require $\tilde{O}\left(\frac{1}{\epsilon^d}\right)$ measurements to cover the space up to radius $\epsilon$. By contrast, an agent that explores this space efficiently can resolve its uncertainty in only $\tilde{O}(d)$ measurements. Thompson sampling with a linear model naturally recovers this performance [57]. The fact that this failure can arise even in a linear system, and even when the density can be estimated precisely, suggests that count-based exploration is not *in general* an effective method for simultaneous exploration with generalization; even if it may be effective at some specific tasks.

### C.5   Ensembles without priors

This paper builds upon a line of research that uses an ensemble of trained models to approximate a posterior distribution. Compared to previous works, our main contribution is to highlight the importance of a 'prior' mechanism in ensemble uncertainty. Figure 13 presents an extremely simple example of 1D regression with a (20,20)-MLP and rectified linear units. The data consists of $x_i = \frac{i-5}{5}$ for $i=0,..,10$ and $y_i = 5\mathbb{1}\{i=10\}$.

The results above highlight the drawbacks of naive ensembles. A pure ensemble trained from random initializations fits the data exactly and leads to almost zero uncertainty anywhere in

Figure 13: Posterior predictive distributions for ensemble uncertainty.

the space [31]. A bootstrapped ensemble takes the variability of the data into account and thus has a wide predictive uncertainty as $x$ grows large and positive. However, where the data has target value zero, bootstrapping will always produce a zero target and consequently the ensemble has almost zero predictive uncertainty as $x$ becomes large and negative [47].

This lack of prior uncertainty can lead to arbitrarily poor decisions, as outlined in [50]. If an agent has only ever observed zero reward, then no amount of bootstrapping or ensembling will cause it to simulate positive rewards. This issue is easily remedied by the addition of a prior mechanism, either through $l_2$ regularization to initial random weights (4), or the addition of a fixed additive random 'prior network' (5).

## C.6 Summary

We summarize the issues raised in Section 2 in Table 1. This table is meant only as a rough summary and should not be taken as rigorous statement. Roughly speaking, a green tick means success, red cross means failure and a yellow circle means something in between. This paper proposes a combination of bootstrap sampling with prior function as an effective computational approximation to Bayesian inference in deep RL. Although our method is somewhat computationally expensive, since it requires training an ensemble of models instead of one, this computation can be done in parallel and so is amenable to large scale distributed computation.

Table 1: Important issues in posterior approximations for deep reinforcement learning.

| | Data conc. | Learned metric | Multi step | Works in noise | Prior effect | Cheap compute |
|---|---|---|---|---|---|---|
| Dropout [20] | ✗ | ✓ | ✗ | ● | ✗ | ✓ |
| NoisyNet [17] | ● | ✓ | ✗ | ✓ | ✗ | ✓ |
| BBB / VI [11] | ● | ✓ | ✗ | ✓ | ● | ✓ |
| Density count [6] | ✓ | ✗ | ✓ | ✗ | ● | ✓ |
| 'Distributional' RL [8] | ✗ | ● | ✓ | ● | ✗ | ✓ |
| Ensemble [31] | ✗ | ✓ | ✓ | ✗ | ✗ | ✗ |
| Bootstrap [47] | ✓ | ✓ | ✓ | ✓ | ✗ | ✗ |
| **Bootstrap + prior** | ✓ | ✓ | ✓ | ✓ | ✓ | ✗ |
| Exact Bayes | ✓ | ✓ | ✓ | ✓ | ✓ | ✗✗✗ |

## Footnotes

[6] This corrects an errant derivation in [46], but maintains the same overall message.

[7] Note that this lower bound is very conservative and provided only for illustration. A more precise analysis would show poor performance even for large $d$.

[8]Any method for Bayesian RL might reasonably claim to be a distributional perspective on reinforcement learning. For this reason, we use quotation marks when we want to distinguish the specific form of distributional RL popularized by [6].

[9]Reward $r_t(x_t) = x_t^T \theta^* + \epsilon_t$ for some $\theta^* \in \mathbb{R}^d$ and $\epsilon_t \sim N(0,1)$ [56].