[Reviews · NeurIPS 2018]

Reviewer 1



Summary: This paper argues for the use of randomized prior functions for exploration in deep reinforcement learning. The authors show that some of the existing approaches do not satisfy some basic sanity checks even in a linear setting. They then show that randomized prior functions do satisfy these sanity checks in a linear setting, and present experimental results on simple simulated domains to argue that the intuition behind these results extends to the non-linear setting. Quality: My main concern with this paper is not very self contained. Many important details are pushed to the appendix (including a fully spelled out algorithm), and/or the reader is pointed to referenced papers to understand this paper. Clarity: While the paper is well structured and easy to read on a macro level, many details have been omitted and/or pushed to the appendix which breaks the logical flow and required me to re-read many paragraphs. Originality: Although randomized prior functions are not new, this paper presents new arguments regarding the logical inconsistencies of some competing methods, and results on the success of randomized prior functions for exploration in RL. Significance: I believe that the algorithms and arguments presented in this paper will inform future research, not only theoretical but also RL applications.

Reviewer 2



Summary: This paper studies RL exploration based on uncertainty. First, they compare several previously published RL exploration methods and identifying their drawbacks (including illustrative toy experiments). Then, they extend a particular previous method, bootstrapped DQN [1] (which uses bootstrap uncertainty estimates), through the addition of random prior functions. This extension is motivated from Bayesian linear regression, and transferred to the case of deep non-linear neural networks. Experimental results on the Chain, CartPole swing-up and Montezuma Revenge show improved performance over a previous baseline, the bootstrapped DQN method. Strength: See conclusion. Comments: - Lemma 2: The lack of uncertainty propagation in standard supervised learning methods is an important point indeed. However, you specifically introduce it as a problem of variational inference, while I think it is a problem that applies to all uncertainty methods from supervised learning used in RL, when their output distribution and loss only concern a mean. The problem indeed is that one needs to explicitly learn/propagate entire distributions or higher-order moments, i.e. specifying additional network outputs apart from the mean, as you mention in the appendix (lines 527-529). I think [5] is an early reference that identified this problem. I don’t think the bootstrap method used in this paper is robust against this problem either actually. If I sample many 1-step bootstrap predictions for a particular state-action pair (and do sample the uncertainty at the next state), then for many samples the bootstrap datasets will become similar, and the entire ensemble starts to fit the same model (uncertainty becoming zero). This is because the loss in bootstrapped DQN still only considers/learns the mean of the distribution which you sample from. - Figure 2: Why is the posterior over x^t*theta almost only varying in the x1 direction? The observed datapoints seem to have equal amount of spread in both the x1 and x2 dimensions. - Lemma 3: This is a key step in your method. I would prefer a derivation of both eq 4 and 5 in the appendix, showing how they lead to the posterior of eq 3. - Line 196: At first it appeared strange to me that the random prior is fixed throughout learning, as for example many previous approaches resample parameter space noise [4] at the beginning of every episode. It might be good to identify that this choice does follow from Lemma 3, i.e. for each sample \tilde{theta} from the prior you get a sample from the posterior, and each bootstrap network covers one such sample from the posterior. - Section 4.1: Are the prior networks also Glorot initialized (and then rescaled by beta)? Or what other distribution do you draw these from? - Appendix B&C: (Chain results) I do not understand what parameter p denotes here. In Lemma 1 and Figure 12, it is the dropout probability. Line 608-609 mentions it as well? - Line 270: … and the continuous action space? - Figure 6: You seem to tune hyperparameters for your BSP, but not for the BS method? I think it would also be good to state how you compare to other directed exploration methods on Montezuma, e.g. [2,3] come to 3500 and 3000 score, respectively (which your best result is close to). Appendix: - Lemma 5: I think this needs a proof/derivation of the lemma? - Line 584: ‘where the data has target value zero, bootstrapping will always produce a zero target’ - - > Is this really true? The bootstrapping procedure is oblivious of the numeric range of the data right? I wonder whether the effect in Figure 9 would also be visible if your left 9 datapoints had y=1 instead of y=0. I think what you are looking at here is the extrapolation of a line when all measured datapoints nearby have the same y. It turns out that naive bootstrap methods then apparently have problems, but I don’t think this has to do with the fact that the y values are specifically 0. (This specific example is still relevant, as in sparse reward domains you will indeed see many returns of 0.) - Table 1: I think it would help to caption the column items, as some of these names were not explicitly mentioned in the text. For example, ‘learned metric’ at first confused me, but checking the crosses and reading back it probably refers to the discongriuity between new states and value uncertainty. -Table 1: ‘multi-step’ means whether uncertainty is propagated? I don’t think that is explicitly done for ensemble/bootstrap methods either right? Or do you mean that you sample your next state prediction Q(s’,a’) from the bootstrap ensemble distribution? But that does not prevent premature convergence of uncertainty, as the bootstrap distribution will still be fitting the mean. - Appendix B: ‘Algorithm 14’ - - > numbering mistake? Conclusion: At first, the proposed modification of random prior functions appeared simple, reminding of random parameter space noise. However, it is more involved and has a deeper theoretical motivation from Bayesian linear regression, and turns out to clearly work better than previous bootstrap ensemble methods in RL. Moreover, I really like the effort in Section 2 and Appendix A to systematically discuss previous attempts with respect to uncertainty in RL (although I think the lack of uncertainty propagation identified for varimethods is actually still present in the current paper). I think the paper could be clarified at a few points (as mentioned in the comments above), such as a derivation of Lemma 3 which is a key step in the paper, and the experimental section could mention results of other methods than the bootstrap DQN paper only. Overall, the paper contributes interesting insights and a novel, simple extension to the statistical bootstrap, which actually would be of interest to NIPS attendees outside RL (e.g., from Bayesian deep learning) as well. [1] Osband, Ian, et al. "Deep exploration via bootstrapped DQN." Advances in neural information processing systems. 2016. [2] Bellemare, Marc, et al. "Unifying count-based exploration and intrinsic motivation." Advances in Neural Information Processing Systems. 2016. [3] Ostrovski, Georg, et al. "Count-Based Exploration with Neural Density Models." International Conference on Machine Learning. 2017. [4] Plappert, Matthias, et al. "Parameter space noise for exploration." arXiv preprint arXiv:1706.01905 (2017). [5] Dearden, Richard, Nir Friedman, and Stuart Russell. "Bayesian Q-learning." AAAI/IAAI. 1998. After rebuttal: The authors elaborated on how their approach does propagate uncertainty, since each bootstrap network only uses itself as a target. I agree that this will propagate uncertainty. I think it is crucial to stress this aspect of their method, around Lemma 2 and probably somewhere in the introduction or discussion, as the authors indicate themselves in the rebuttal as well.

Reviewer 3



I enjoy the paper very much as it carefully discusses the problems with uncertainty estimate and its (sometimes underestimated role) in deep RL. Summary: The paper starts with previous methods on uncertainty estimate and their applications to RL. By noting that these previous methods all have disadvantages, the paper proposes a prior mechanism for exploration, which passes sanity check under linear representation and is shown to over-perform deep RL algorithms without a prior mechanism. Questions: 1. In line 68, why should exploration be to prioritize the best experiences to learn from. Probably the author can elaborate more on this concise statement. 2. Lemma 1 seems a bit vague and too general. Though I could clearly see the reason why the posterior produced by dropout does not concentrate. Probably more explanation will help. 3. What exactly is the confounding effect of sgd optimization mentioned in line 117? 4. Lemma 2: why the objective is a variational loss? Does it follow from the variational objective of one variational distribution against a posterior, what are these? 5. I am familiar with the work of distributional RL, and I understand the orthogonal role of distributional RL and Bayesian prior. However, I do not completely understand the figures in Figure 1. The author should explain the experiments and illustrations more clearly. 6. How are the graphs in Figure 2 obtained? Can authors explain a bit more in the paper? It is a bit compounding at the first sight. 7. In line 214, does a random neural network refer to a network with random weight initializations? One per each ensemble member? 8. In Figure 5, the initial performance of BSP (light blue) is obviously worse than the other algorithms especially when zooming in in (b). What is the reason? General Advice: 1. This is no doubt a paper with rich content, and can probably use better presentation. The paper presents a lot of topics related to uncertainty estimates (in RL and deep learning) and their corresponding weaknesses. However, after reading up to page 5, I have very little idea as to how these various methods relate to the proposed prior mechanism. It feels a bit like reading a list of methods without too much contrast. Though the authors have definitely identifies their disadvantages using small and contrived experiments. I just feel that the first five pages can be better organized and presented.